# The Human Side of Leadership: Exploring the Impact of Servant Leadership on Work Happiness and Organizational Justice

**DOI:** 10.3390/bs14121163

**Published:** 2024-12-04

**Authors:** Jesus Alberto Agustin-Silvestre, Miluska Villar-Guevara, Elizabeth Emperatriz García-Salirrosas, Israel Fernández-Mallma

**Affiliations:** 1Escuela Profesional de Administración, Facultad de Ciencias Empresariales, Universidad Peruana Unión, Lima 15102, Peru; albertoagustin@upeu.edu.pe; 2Escuela Profesional de Administración, Facultad de Ciencias Empresariales, Universidad Peruana Unión, Juliaca 21100, Peru; 3Faculty of Management Science, Universidad Autónoma del Perú, Lima 15842, Peru; egarciasa@autonoma.edu.pe; 4Escuela Profesional de Ingeniería Civil, Facultad de Ingeniería y Arquitectura, Universidad Peruana Unión, Juliaca 21100, Peru; pastorisrael@upeu.edu.pe

**Keywords:** promoting leadership, servant leadership, work happiness, organizational justice, Peru

## Abstract

The leadership literature suggests that a servant leadership style can reduce negative employee outcomes, even in challenging work environments such as the educational sector, where teachers play a key role in social development. This research aimed to evaluate the effect of servant leadership on work happiness and organizational justice. An explanatory study was carried out including 210 men and women who declared that they perform teaching activities, aged between 21 and 68 years (M = 38.63, SD = 10.00). The data were collected using a self-report scale of servant leadership, work happiness and organizational justice, obtaining an adequate measurement model (α = between 0.902 and 0.959; CR = between 0.923 and 0.963; AVE = 0.604 and 0.631; VIF = between 1.880 and 2.727). The theoretical model was evaluated using the Partial Least-Squares PLS-SEM method. According to the results, the hypotheses were confirmed, demonstrating that there is a significant positive effect of servant leadership on work happiness (β = 0.69; *p* < 0.001) and organizational justice (β = 0.24; *p* < 0.001) and a positive effect of work happiness on organizational justice (β = 0.61; *p* < 0.001). This research provides valuable insight for educational leaders seeking to improve perceptions of happiness and justice in their organizations and promotes servant leadership to achieve this goal.

## 1. Introduction

Leadership studies have addressed an important spectrum of styles and their benefits for diverse organizations [1,2,3,4,5,6]. Some of these styles can be considered relatively recent, such as servant leadership [7,8]. This is a construct that establishes that the leader, before assuming his role, must be a servant [9,10,11]. That is, you must serve your work group first and, as a natural response, they will take you in as their leader [12,13]. This leadership style stands out from the rest because it facilitates and encourages people to reach a high level of commitment to their organization, improving their performance, due to the treatment they receive from their leader [14,15]. Leaders who practice servant leadership ensure that their work groups develop their career paths and even take an interest in their physical well-being [16]. Likewise, the positive effects of the servant leader include greater appreciation and respect for one’s own organization and higher levels of acceptance of leadership, and they are also associated with voice behavior and spirituality [17]. Studies have shown that this leadership style is effective in alleviating depressive symptoms in populations in the educational sector [18].

Although studies have been carried out on the impact of leadership in different areas of knowledge, there are two variables that have achieved an important place in the scientific literature: work happiness and organizational justice. Work happiness is a highly valued aspect in organizations [19] because it implies greater performance and is essential to live a healthy life and build a better partnership between the employee and the organization [20]. On the other hand, maintaining positive employee attitudes is a huge challenge because it requires high levels of resources and energy. The study by Gonzales-Macedo et al. [21] found significant effects of servant leadership on emotional salary and indirect effects on work happiness. Employees with high rates of workplace happiness have shown better employee citizenship behaviors [22]. Additionally, there are studies that support a positive relationship between work happiness and internal communication [23]. Moreover, studies have shown a positive impact of servant leadership on organizational justice, and it has reducing effects on negative work outcomes for employees [9]. Although there is scarce research on the matter, studies have shown that servant leadership and organizational justice have a positive impact on the behavior of organizational citizenship [24].

Thus, after a review of the aforementioned antecedents, the inclination to discern these topics among education professionals and academics is evident. Bibliometric indicators show the ten countries with the greatest dissemination of scientific results: the United States, China, United Kingdom, Pakistan, Turkey, Malaysia, India, Australia, Spain and Indonesia. These are the same countries that have mostly worked in diverse populations, areas and sectors, such as business, social sciences, psychology, economics and medicine. On the other hand, when searching out scientific dissemination by country, research regarding the subject conducted in the Peruvian population was not found, limiting scientific support and guidance for future studies in educational contexts. Furthermore, there is no evidence of any preceding empirical study that has delved into the behavior of these variables as a whole. In this sense, the objective of this research was to evaluate the effect of servant leadership on work happiness and organizational justice.

The present investigation is divided into the following sections: Section 2 contains the literature review. Section 3 provides the materials and methods. Section 4 focuses on the results. Section 5 describes the discussion and Section 6 the conclusions of this research.

## 2. Literature Review

### 2.1. Servant Leadership

Research in the field of leadership is diverse and has addressed the different models or styles of leadership, based on the various theories raised since the beginning of the 20th century [25]. Both empirical and psychometric studies cover leadership models, such as ethical [26,27], authentic [28,29], servant [7,30,31,32], transactional [33], situational [25], transformational [33], democratic [25], autocratic [34], laissez faire [35], strategic [36], bureaucratic [37], charismatic [38], people-oriented [38], natural [39], task-oriented [40] and relationship-oriented models [41]. From the interest in studying a leadership model that focuses on people and their growth, servant leadership emerges as an area of knowledge worthy of further study due to its positive results in the working group [42] and because it also promises to develop in a scenario of high moral and ethical standards centered on people [43]. It was in 1970, with the publication of “The Servant as Leader”, that Robert Greenleaf proposed that the most accurate test of a servant leader is whether they make the conscious decision to be a servant first and, as a natural response, their aspiration to lead is accentuated [44]. Influential leaders are actually those who know how to win the favor of others before asking them to follow their ideals [26].

This model proposes that servanthood should be the distinguishing characteristic of leadership. The exercise of servant leadership would not only make today’s organizations more optimal and better managed, but they would also have stronger organizational principles and become sustainable over time. Greenleaf [44] affirms that business leaders who established this model of leadership in a primogenital and unanimous way, by precept, as an example, would discover a greater purpose that would elevate the aspect of their service and bring joy in the life of the work group. Nearly five decades have passed since these words were first articulated, and the diversity of research has affirmed that servant leadership is gaining more and more followers in university chairs, seed programs, business schools, among executives, in government, and in public and private institutions.

Additionally, according to Espinosa and Esguerra [45], servant leaders have qualities that distinguish them from others and help them advance the achievement of a common good that benefits all parties because they build a specialized approach to influencing others in the accomplishment of their tasks [46]. To achieve established goals and objectives, the servant leader’s primary goal is to serve others by investing in their well-being and progress [47]. Similarly, Pino et al. [30] submit that servant leaders have as their main objective to allow those they lead to develop their full potential in a series of areas, such as the growth of self-motivation and leadership capabilities, as well as the defense of empowerment. Likewise, Jaramillo et al. [48] consider it to be the act of directing workers in the execution of a strategy that facilitates the fulfillment of corporate objectives.

### 2.2. Work Happiness

Workplace happiness is the psychological well-being that an individual employee experiences in a particular work situation [49]. Some have defined it in terms of pleasant experiences (positive feelings and moods) at work [19], although there are discussions regarding how this construct can be measured, which depends on the level at which the experiences are observed, their duration and content, and the benefits received. Empirical studies have shown that work happiness, seen as the well-being of employees, involves three basic aspects: life, work and psychology [50,51,52]. Happiness at work influences job satisfaction and emotions closely related to work, and some associate it with quality of work life [53]. During the COVID-19 pandemic, it was observed that job insecurity had a negative predictive effect on the production of individual work happiness [54,55].

A high level of happiness at work means that employees have positive emotional experiences that motivate work effort, which helps them to be more motivated to handle their work [56]. Greater job happiness leads to better job performance. Improving the conditions for happiness at work means that employees have sufficient emotional and psychological resources to best cope with the demands of their work. As a result, their work engagement will increase [57].

Some specialists formulate three components of work happiness: engagement, job satisfaction and emotional organizational commitment [58]. On the other hand, in the model of Dutschke et al. [59], five components have been considered: achievement of objectives, leadership, sustainability and work–family balance, work group organization, and self-realization [20]. Other theoretical models for work happiness have also been built [52,60]. In contrast to the aforementioned models, there are theoretical proposals that consider two components to define work happiness [61]: factors related to the job position and factors related to the worker, based on Fisher’s theoretical approach [19].

*Factors related to the job*—aspects of the work environment and work conditions that can influence work happiness—are the physical environment, the workload and work–life balance, autonomy and control, interpersonal relationships, and recognition and rewards, although these largely focus on hedonic experiences of pleasure and/or positive beliefs about an object, for example, job satisfaction, affective commitment and the experience of positive emotions while working. On the other hand, *worker-related factors* are intrinsic aspects of employees that affect their perception of work happiness, such as personality and values, skills and competencies, intrinsic motivation, physical and emotional health, and expectations and aspirations.

### 2.3. Organizational Justice

This term refers to the way employees perceive what is fair and unfair in their workplace [62]. That is, workers tend to personally evaluate the ethical and moral standards of their organization. Because justice is a descriptive concept, based on worker observation, this can affect the level of commitment and trust, increase or reduce organizational performance, and even influence the behavior of individuals [63]. The dimensions of this construct are distributive justice, procedural justice and interpersonal justice [61].

Distributive justice includes the individual’s perception of fairness within his or her organization, such as rewards and incentives, remuneration, and promotions. Its emphasis is on benefits and equitable distribution, and it is related to cognitive, affective and behavioral reactions [64]. Procedural justice has to do with the search for equity in the processes that the organization uses to distribute the various benefits among its workers. Attitudes and behaviors resulting from employees’ perceptions of inadequate distribution can generate indignation that translates into resentment and lack of commitment [65]. Interpersonal justice has to do with the interaction that occurs between the leader and those led by the leader and the treatment they receive when they apply the company’s procedures. Furthermore, it can be evidenced in the respect, courtesy and sense of dignity given to an employee [66,67,68].

Based on what was stated in the previous paragraphs, Figure 1 shows graphically the hypotheses of this study, which are detailed below:

**H1.** 
*Servant leadership will have a significant positive effect on work happiness.*


**H2.** 
*Servant leadership will have a significant positive effect on organizational justice.*


**H3.** 
*Work happiness will have a significant positive effect on organizational justice.*


## 3. Materials and Methods

### 3.1. Study Design and Participants

A cross-sectional and explanatory study was designed [69]. The population was made up of Regular Basic Education (EBR) teachers from the Puno region of Peru. Only people who met the following inclusion criteria were included in the study: being engaged in teaching at the time the questionnaire was administered, being affiliated with an educational institution in the Puno region in any type of work and having practiced for a minimum of six months. To define the sample size, non-probabilistic sampling was chosen [70], and the electronic tool Soper was used [71]. This tool takes into account the number of both observed and latent variables in the SEM, along with the anticipated effect size (λ = 0.24), the desired level of statistical significance (α = 0.05) and the required statistical power (1 − β = 0.76). Based on these parameters, the need to include 182 teachers in the sample was determined. However, a total of 210 individuals participated, with an almost equal distribution between women (53.8%) and men (46.2%), with ages ranging between 21 and 68 years (M = 38.63 and SD = 10.00). Table 1 shows that the majority of participants were between 31 and 40 years old (38.1%), single (46.7%), received higher university education (75.2%), worked in the private sector (76.2%) and had between 1 and 5 years of employment as a teacher (68.1%).

### 3.2. Instrument

The questionnaire had a 5-point Likert-type response format, ranging from 1 (strongly disagree) to 5 (strongly agree). In the first section, instructions for filling out the questionnaire were given; the second section requested sociodemographic information in order to determine the profiles of the participants; and in the last section, the measurement scales detailed below were presented:

Concerning the scale to measure work happiness, the metric originally designed by Del Junco et al. [72], which was later translated and validated into Spanish [61], was used. The scale shows two components: (1) factors related to the job and (2) factors related to the worker. It is a brief measure made up of 11 items that evaluate the degree of happiness that the worker has in his or her work environment. An example item is “*I enjoy doing my job well*”. The scale reports optimal values of internal consistency (α = 0.938; CR = 0.947; AVE = 0.620 and VIFs = 2.727).

The scale used to measure servant leadership was designed and validated in South America by Gocen and Sen [43]. This is a short scale with a unidimensional structure that consists of 7 items, an example item being “*My leader prioritizes my interests ahead of his own*”. The scale reaches optimal values of internal consistency (α = 0.902; CR = 0.923; AVE = 0.631 and VIFs = 1.880).

To measure organizational justice, the scale originally developed by Moorman [73] and later used in Spanish [74] was used. It consists of 17 items distributed across 3 factors, an example item being “*Provide opportunities to discuss or appeal to a decision made*”. Distributive justice was measured using 5 items that attempted to capture teachers’ perceptions of the degree to which the educational institution rewards fairly. Procedural justice was measured using 6 items that sought evidence of perceptions of justice in organizational processes. The interpersonal justice dimension consisted of 6 items that assessed whether or not supervisors managed in a respectful and fair manner. The scale reports optimal values of internal consistency (α = 0.959; CR = 0.963; AVE = 0.604 and VIFs = 2.628).

### 3.3. Procedure and Ethical Considerations

The research was previously evaluated and approved by the Ethics Committee of a private university in Peru (2023-CEEPG-00072). Subsequently, during the period from October 2023 to March 2024, a Google form was used, where participants who decided to take part in the study read and completed each item. Prior to data collection, respondents were informed that their participation was voluntary and anonymous. Furthermore, data confidentiality rules and the principles of the Declaration of Helsinki were followed during this research [75,76]. Informed consent was collected from each participant, who gave their consent with the following statement: “*I acknowledge that by completing this questionnaire I am giving my consent to participate in the study*”.

### 3.4. Data Analysis Procedure

The PLS-SEM Partial Least-Squares method was used to statistically analyze the data and test the hypotheses. PLS-SEM is a comprehensive multivariate statistical analysis approach that includes measurement and structural components to simultaneously examine the relationships between each of the variables in a conceptual model, and it is suitable for multivariate analysis, that is, it involves a number of variables equal to or greater than three [77]. Also, this research employed PLS-SEM because it facilitates the development of theories [78]. WarpPLS (version 8.0) was used to perform the PLS-SEM analysis. The reason for using this software is that WarpPLS offers the option of using different algorithms for external and internal models [79], identifying and taking into account non-linear relationships when calculating variable scores such as path coefficients and non-relevant *p*-values [80].

## 4. Results

Evaluating a model by PLS-SEM involves two stages that entail the evaluation of the measurement and structural models, which are detailed below:

### 4.1. Evaluation of the Measurement Model

To evaluate the quality of reflective constructs, the convergent validity and reliability of the constructs, that is, internal consistency, must be evaluated [78,81,82]. And the following indicators must be met (Table 2):

Table 3 shows that all indicators are met. All loadings comply with being greater than 0.70, except for items SL7 and OJ4, whose values are 0.672 and 0.681, respectively. Furthermore, jointly, the constructs provide good indicators, since the Cronbach’s Alpha and CR values are greater than 0.70. Likewise, the AVE results are also sufficient, since all of them are greater than 0.604. The full collinearity VIFs are also acceptable, since all values are less than 2.727, such that they are in the required range. Given that all the indicators were acceptable, the discriminant assessment was carried out.

The degree to which each construct is distinct from the other constructs in the model is determined by discriminant validity [81]. To pass discriminant validity, the square root of the AVE value of each construct must be greater than the highest correlation between the construct and the other constructs in the model [78,79,81]. In that sense, this research was shown to comply with the condition, that is, the model has an acceptable discriminant validity (see Table 4).

### 4.2. Structural Model Evaluation

In order to evaluate the structural model, two preliminary criteria were verified and reported: the significance of the path coefficients and the values of the R^2^ coefficients for the endogenous constructs. Each of the hypotheses raised were causally linked in the structural model—a design that represents the relationship between a pair of constructs. Path coefficients and their *p*-values were calculated for each relationship in the model. Although each path coefficient should be significant, the value of the R^2^ coefficient is highly dependent on the field of study. Chin [83] suggests values of 0.67, 0.33 and 0.19 as, respectively, substantial, moderate and weak measures of R. In behavioral studies, a value of 0.2 for R^2^ is generally considered acceptable [84,85].

In the present study, the R^2^ values for the WH and OJ coefficients were 0.47 and 0.62, respectively. Therefore, all R^2^ values had relatively high and acceptable values. The values from this study suggest that the variables account for a high percentage of the variance in OJ.

Table 5 and Figure 2 show the results of the hypothesis tests and the evaluation of the path coefficients. The results show the significant positive effects of SL on WH (H1), SL on OJ (H2) and WH on OJ (H3). In this way, the three hypotheses were tested, and these results highlight the importance of servant leadership as a key factor in promoting work happiness and staff well-being. Also, the importance of servant leadership in promoting organizational justice and its associated benefits, such as trust, equity, commitment and talent retention, are highlighted. This has important implications for human resource management, leadership development and overall organizational culture. Furthermore, it is highlighted that work happiness and organizational justice are closely interrelated and that promoting happiness at work can have positive effects on the perception of justice within an organization, which in turn can generate additional benefits in terms of commitment, productivity and retention of collaborators.

For the fit index of the global model, the six goodness-of-fit indices were considered [79], with a confidence level of 95%. The efficiency indices are as follows:Average trajectory coefficient (APC) and *p* < 0.05;Average R-squared (ARS) and *p* < 0.05;Adjusted R-mean square (AARS) > 0.02 and *p* < 0.05;Block average VIF (AVIF): acceptable if ≤5, ideally ≤ 3.3;Average complete collinearity (AFVIF): acceptable if ≤5, ideally ≤ 3.3;Tenenhaus GoF (GoF): small ≥ 0.1, medium ≥ 0.25, large ≥ 0.36.

For this investigation, the six fit indices suggested that the model fit was more than acceptable: average path coefficient (APC) = 0.512, *p* < 0.001; average R^2^ (ARS) = 0.547, *p* < 0.001; adjusted average R^2^-squared (AARS) = 0.544, *p* < 0.001; average block variance inflation factor (AVIF) = 1.816 (acceptable if ≤5, ideally ≤ 3.3); average full collinearity variance inflation factor (AFVIF) = 2.412 (acceptable if ≤5, ideally ≤3.3); and Tenenhaus GoF (GoF) = 0.582 (small ≥ 0.1, medium ≥ 0.25, large ≥ 0.36). The predictive validity of a construct is confirmed when the value of its linked R^2^ coefficient is greater than zero. This being the case with the values of the endogenous variables of the model translates into an acceptable predictive validity throughout the model.

## 5. Discussion

The objective of this research was to evaluate the effect of servant leadership on work happiness and organizational justice. The results provide evidence in favor of the structural model originally proposed, and the application of this research suggests a significant contribution to the promotion of the good practice of servant leadership as an indispensable factor in groups, societies and institutions [46,86,87,88,89] and that it also has an impact on factors related to the work life of Peruvian teachers. This precedent suggests the continuation of more in-depth research to evaluate the behavior of new approaches to humanized leadership, reporting robust indicators regarding the management of human talent. According to the results, it has been shown that servant leadership will have a significant positive effect on happiness at work. The practice of inclusive leadership with fair practices can have a significant impact on employee happiness, as demonstrated by Jha et al. [90], which supports the result obtained. Hamid [91] has shown that the servant leadership style is optimal for creating happiness at work because it stimulates a positive attitudinal state in employees.

Likewise, it has been found that servant leadership had a significant positive effect on organizational justice. This result is perfectly consistent with previous studies that have reported the positive perception of workers who admit higher indicators of justice in their workplace when supervisors or immediate bosses apply more humanized and people-oriented leadership styles, allowing the area of human talent management to exceed the expectations of organizational behavior [10,24,92]. Studies have also been reported showing a positive relationship between these variables [9,93].

Evidence from this research establishes the positive effect of job happiness on organizational justice. Although few studies have demonstrated this link, previous publications have suggested a possible influence between both constructs, recommending their inclusion in future research on leadership and organizational justice. In that sense, the study by Jha et al. [90] strongly suggests that an inclusive leadership style which considers the contribution and participation of the group would have a connection with the happiness of the work group; thus, organizational leaders who promote inclusion and teamwork may be the key to organizational growth and development.

On the other hand, other research has reported findings that, in turn, address the three variables used in this study, focusing on the importance of future research in challenging sectors such as the teaching population [21,90]. While this research suggests a positive relationship between servant leadership and workplace happiness, it is critical to consider that other factors such as organizational learning, organizational facilitators and affective commitment could have some degree of influence [20,29]. Future studies could use variables such as spiritual leadership or transformational leadership to observe the behavior of these constructs and make comparisons [94,95]. On the other hand, servant leadership can be effectively implemented in diverse and often challenging educational settings, such as the Puno region [96,97]. What specific strategies might educational leaders use to develop a more service-oriented approach, and how might these strategies be adapted to local needs and contexts? These discussions highlight the importance of an integrative and reflective approach to understanding and applying servant leadership principles to enhance workplace happiness and organizational justice in a diverse context.

### 5.1. Theoretical and Managerial Implications

From a theoretical perspective, this study contributes to the understanding of servant leadership, happiness at work and organizational justice, deepening each approach and providing a theoretical precedent that could be the beginning of other more complex models. Likewise, its usefulness in the educational context of such a challenging Peruvian region as Puno, where the application of scientific studies on this topic is limited due to cultural, political and educational conditions, could suggest a valuable contribution to the informed decision making of educational workers. When analyzing the results, the implication of implementing leadership styles that are more focused on people and less egocentric, such as servant leadership, is denoted; in that sense, area chiefs and directors of educational institutions could redefine the characteristics and descriptions of trust and strategic positions so that, through top management, servant leadership can be disseminated and its benefits can be made real in the work group. In addition, the business impact is important because it provides valuable information to educational leaders and relevant authorities in Peru, especially in the Puno region, allowing them to design policies and management practices that promote a more satisfactory and fair working environment for teachers, which, in turn, can improve the quality of education in the region and transfer that influence to the whole country.

### 5.2. Limitations and Future Research

Although this research contributes an important legacy to science, the methodological and contextual limitations must be recognized, and, with these, the need to conduct more research to confirm and expand these findings in other populations and educational settings. Although 210 teachers participated in this study, the Puno region may have a very diverse educational and cultural environment. Therefore, the results may not be generalizable to other educators in different geographic, cultural or socioeconomic contexts, and the sample may not have been large enough to capture all the perceptions in the sector.

Self-report surveys in the data collection process require trust in the honesty and accuracy of each participant’s responses. However, teachers can react in a biased way to give a more favorable image of themselves or their working conditions, which can distort the results and their interpretation, which is why it is recommended to investigate other groups of teachers in turn. It is suggested to add focus groups and in-depth interviews to corroborate the greater proximity to the reality experienced by the sector.

Although this research focused on servant leadership and its impact on workplace happiness and organizational justice, other uncontrolled variables may also have influenced the results. For example, personal factors such as teachers’ past work experience or personality can influence their perceptions of work happiness and organizational justice. Therefore, it is recommended to consider these factors in future studies.

## 6. Conclusions

The leadership literature suggests that a servant leadership style can reduce negative employee outcomes, even in challenging work environments such as the Peruvian education sector, where teachers play a leading role in social development. Therefore, this study set out to evaluate the effect of servant leadership on workplace happiness and organizational justice. In this sense, to address the main objective of the research, an explanatory study was carried out considering 210 men and women who declared that they perform teaching activities, aged between 21 and 68 years (M = 38.63 and SD = 10.00). The theoretical model was evaluated using the Partial Least-Squares PLS-SEM method. The hypotheses were confirmed, demonstrating the positive effect of servant leadership on work happiness (β = 0.69; *p* < 0.001) and organizational justice (β = 0.24; *p* < 0.001), as well as the effect of work happiness on organizational justice (β = 0.61; *p* < 0.001). This research provides valuable insight for educational leaders seeking to improve perceptions of happiness and justice in their organizations and promotes servant leadership to achieve this goal. Therefore, this study has important implications for education managers, human talent personnel and academic professionals.

## Figures and Tables

**Figure 1 behavsci-14-01163-f001:**
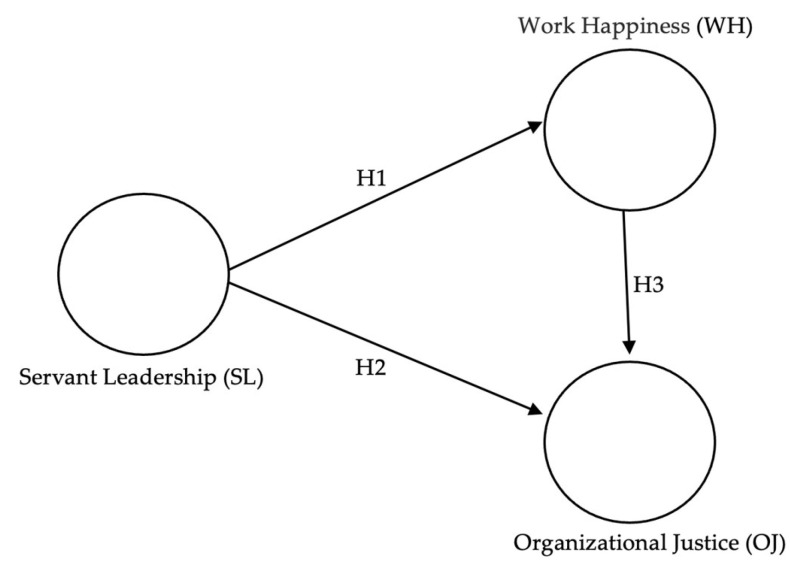
Hypothetical model.

**Figure 2 behavsci-14-01163-f002:**
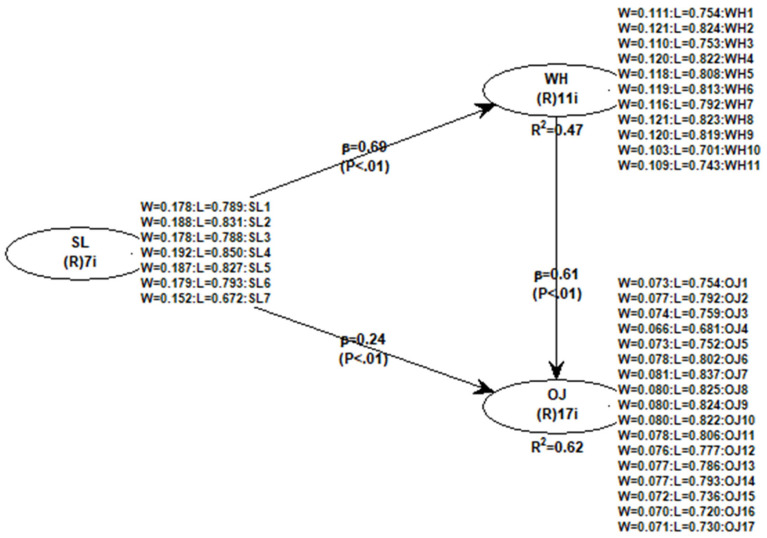
Structural model results.

**Table 1 behavsci-14-01163-t001:** Sociodemographic characteristics of the participants (n = 210).

Characteristic	Category	Frequency	%
Sex	Female	113	53.8
	Male	97	46.2
Age range	21–30 years	54	25.7
	31–40 years	80	38.1
	41–50 years	50	23.8
	51–68 years	26	12.4
Civil status	Single	98	46.7
	Married	88	41.9
	Cohabitant	18	8.6
	Divorced	4	1.9
	Widower	2	0.9
Greater academic instruction	Technical superior	18	8.6
	University higher	158	75.2
	Master’s degree	34	16.2
Laboral sector	Private	160	76.2
	Public	50	23.8
Years of teaching service	1 to 5 years	143	68.1
	6 to 10 years	30	14.3
	11 to 15 years	30	9.5
	More than 15 years	17	8.1

**Table 2 behavsci-14-01163-t002:** Indicators used to evaluate the convergent validity and reliability of the constructs.

Indicator	Level
Loading (L)	>0.7
Composite reliability (CR)	>0.7
Cronbach’s alpha (α)	>0.7
Mean-variance extracted (AVE)	>0.5
Variance inflation factor (VIF)	<5
Significance level (*p*-value)	<0.05

**Table 3 behavsci-14-01163-t003:** Results of the evaluation of the measurement model.

Predictor	Item	Loading	*p*-Value	α	CR	AVE	Full Collinearity VIFs
Servant Leadership (SL)	SL1	0.789	<0.001	0.902	0.923	0.631	1.880
	SL2	0.831	<0.001				
	SL3	0.788	<0.001				
	SL4	0.850	<0.001				
	SL5	0.827	<0.001				
	SL6	0.793	<0.001				
	SL7	0.672	<0.001				
Work Happiness (WH)	WH1	0.754	<0.001	0.938	0.947	0.620	2.727
	WH2	0.824	<0.001				
	WH3	0.753	<0.001				
	WH4	0.822	<0.001				
	WH5	0.808	<0.001				
	WH6	0.813	<0.001				
	WH7	0.792	<0.001				
	WH8	0.823	<0.001				
	WH9	0.819	<0.001				
	WH10	0.701	<0.001				
	WH11	0.743	<0.001				
Organizational Justice (OJ)	OJ1	0.754	<0.001	0.959	0.963	0.604	2.628
	OJ2	0.792	<0.001				
	OJ3	0.759	<0.001				
	OJ4	0.681	<0.001				
	OJ5	0.752	<0.001				
	OJ6	0.802	<0.001				
	OJ7	0.837	<0.001				
	OJ8	0.825	<0.001				
	OJ9	0.824	<0.001				
	OJ10	0.822	<0.001				
	OJ11	0.806	<0.001				
	OJ12	0.777	<0.001				
	OJ13	0.786	<0.001				
	OJ14	0.793	<0.001				
	OJ15	0.736	<0.001				
	OJ16	0.720	<0.001				
	OJ17	0.730	<0.001				

**Table 4 behavsci-14-01163-t004:** Discriminant validity.

	SL	WH	OJ
SL	**0.795**		
WH	0.651	**0.788**	
OJ	0.634	0.767	**0.777**

Note: The square root of AVEs is shown diagonally in bold.

**Table 5 behavsci-14-01163-t005:** Hypothesis testing results.

H	Hypothesis	Path Coefficient	*p*-Value	Decision
H1	SL-WH	0.688	<0.001	Accepted
H2	SL-OJ	0.240	<0.001	Accepted
H3	WH-OJ	0.607	<0.001	Accepted

## Data Availability

Data availability can be requested by writing to the corresponding author of this publication.

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
