# Peer review of "The Human Side of Leadership: Exploring the Impact of Servant Leadership on Work Happiness and Organizational Justice"

_behavsci, 2024, doi:10.3390/bs14121163_

Round 1
Reviewer 1 Report
Comments and Suggestions for Authors
Dear author(s) of: The human side of leadership.
Thank you for providing this study. I have read it with interest and have a few comments that hopefully will be useful for you to further improve the paper.
1. The manuscript should be copy-edited – it contains a few errors and some very long sentences. That should have been improved.
2. The introduction is ok, but a clearer formulation of what constitute the unique contribution of the study to existing knowledge both locally and globally might be added.
3. The theory review is short and might be expanded to give a broader background to the study discussion.
4. The discussion is somewhat shallow, not penetrating deeper into details of the study findings. The implications and managerial implications are also just touched upon and might be made more detailed and illuminating.
Hopefully, the comments may be useful for your effort to improve the paper. I wish you good luck with work.
Comments on the Quality of English Language1. The manuscript should be copy-edited – it contains a few errors and some very long sentences. that should have been improved.
Author Response
|
Thank you for providing me with this study. I have read it with interest and have some comments which I hope you will find useful in improving the article. The manuscript should be corrected: it contains some errors and some very long sentences. That ought have corrected . |
Dear reviewer, thank you for your comments. The English has now been corrected to perfect the final version of the manuscript. |
|
The introduction is fine, but a clearer formulation of what constitutes the study's unique contribution to existing knowledge at both local and global levels could be added. |
Dear reviewer, thank you for your comments. The contribution of the research has now been made clearer. See lines 67-71. |
|
The review of the theory is brief and could be expanded to give a broader context to the discussion of the study. |
Dear reviewer, thank you for your comments. The theory has now been expanded. See lines 70-103, 125-129, and 334-354. |
|
The analysis is somewhat superficial and does not go into the details of the study's results. The implications and implications for management are also briefly mentioned and could be more detailed and enlightening. I hope you find the comments useful in improving the article. I wish you the best of luck with your work.
|
Dear reviewer, thank you for your valuable comments. It has already been corrected following your recommendation in the section “ Theoretical and managerial implications” . See lines 353-370. |

Reviewer 2 Report
Comments and Suggestions for Authors
The paper considers very interesting topic focused on the specific type of leadership. There are lots of papers dealing with different aspects of leadership and leadership styles, however there is a gap when it comes to explore impact within different organizational, national or cultural context. Authors analyzed effects of servant leadership on work happiness and organizational justice in the Puno region, Peru. Survey was conducted among Regular Basic Education teachers.
Research design, hypothesis and methods applied in the paper are clearly stated, arguments are balanced, and results are clearly presented. Conclusions are consistent with the evidence and arguments presented. Cited references are relevant and novel. Authors addressed important questions and conclusions could be interesting for the readership of the journal.
The paper could be improved in the Literature review (Section 2.1) by elaborating more on leadership styles, advantages and disadvantages of servant leadership comparing to other leadership styles. Why authors chose servant leadership to evaluate impact on work happiness and organizational justice and not some other leadership style? Elaboration of previous question could be interesting for possible readers, especially in cases when lots of organizational and policy changes in education have happened.
Author Response
|
El artículo aborda un tema muy interesante centrándose en el tipo específico de liderazgo. Existen muchos artículos que tratan diferentes aspectos del liderazgo y los estilos de liderazgo, sin embargo, existe una brecha cuando se trata de explorar el impacto dentro de diferentes contextos organizacionales, nacionales o culturales. Los autores analizaron los efectos del liderazgo de servicio en la felicidad laboral y la justicia organizacional en la región de Puno, Perú. La encuesta se realizó entre docentes de Educación Básica Regular. |
Estimado revisor, gracias por sus comentarios. En efecto, este estudio ha sido desarrollado con el propósito de abordar temas de interés para el público empresarial actual. |
|
El diseño de la investigación, la hipótesis y los métodos aplicados en el artículo están claramente enunciados, los argumentos son equilibrados y los resultados se presentan con claridad. Las conclusiones son consistentes con la evidencia y los argumentos presentados. Las referencias citadas son relevantes y novedosas. Los autores abordaron cuestiones importantes y las conclusiones podrían ser interesantes para los lectores de la revista. |
Estimado revisor, gracias por sus comentarios. En verdad, este equipo de investigación ha puesto mucho esfuerzo en cumplir con las expectativas de esta prestigiosa revista. |
|
El artículo podría mejorarse en la revisión de la literatura (Sección 2.1) profundizando en los estilos de liderazgo, las ventajas y desventajas del liderazgo de servicio en comparación con otros estilos de liderazgo. ¿Por qué los autores eligieron el liderazgo de servicio para evaluar el impacto en la felicidad laboral y la justicia organizacional y no otro estilo de liderazgo? Desarrollar la pregunta anterior podría ser interesante para los lectores potenciales, especialmente en los casos en que ha habido muchos cambios organizacionales y de políticas en la educación.
|
Estimado revisor, gracias por sus comentarios. La sección 2.1 ha sido mejorada teniendo en cuenta su contribución. Ver líneas 80-104. |

Reviewer 3 Report
Comments and Suggestions for Authors
First, I want to congratulate you on a very well written paper. However, I have some comments:
- at the end of the introduction section, you should present a brief structure of your paper;
- the literature review is very well written, adequately describing the present state of the knowledge in the field of servant leadership;
- lines 140-142: the hypotheses should clearly state the type of influence (direct/indirect; positive/negative)
- the sample is very well described, together with descriptive statistics, as well the way of chosing the sample size (Soper)
- the survey is adequately presented, including the sources for the scales
- the statistical method of analysing the data (PLS-SEM) is adequately presented and justified
- the results section is comprehensive and shows the statistical indicators for the analysed data
- in my opinion you could expand the discussion section, discussing each of the three hypothesis, including theoretical, practical and policy implications as well as their alignment with other similar studies which agree or contradict your findings
Author Response
|
- First of all, I would like to congratulate you on a very well-written article. However, I have a few comments: - at the end of the introduction section, you should present a brief structure of your article;
|
Dear reviewer, thank you for your comments. It has now been corrected. Please see lines 73-76. |
|
- The literature review is very well written and adequately describes the current state of knowledge in the field of servant leadership.
|
Dear reviewer, thank you for your comments. The research team has worked hard to produce quality work that meets the expectations of this journal. |
|
- Lines 140-142: Hypotheses must clearly indicate the type of influence (direct/indirect; positive/negative)
|
Dear reviewer, thank you for your comments. It has now been corrected. Please see lines 168-170. |
|
- The sample is very well described, along with the descriptive statistics, as well as how to choose the sample size (Soper)
|
Dear reviewer, thank you for this comment that commits us to continue contributing with quality. |
|
- The survey is adequately presented, including the sources of the scales.
|
Dear reviewer, thank you for this kind comment. |
|
- The statistical method of data analysis (PLS-SEM) is adequately presented and justified.
|
Dear reviewer, thank you for this kind comment. |
|
- The results section is complete and shows the statistical indicators of the data analyzed.
|
Dear reviewer, the research team thanks you for taking the time to review this manuscript and for your kind words. |
|
- In my opinion, you could expand the discussion section, discussing each of the three hypotheses, including theoretical, practical and policy implications, as well as their alignment with other similar studies that agree or contradict their findings.
|
Discussion ” section has been improved taking into account your contribution. See lines 334-354. |

Round 2
Reviewer 1 Report
Comments and Suggestions for Authors
Dear author(s),
Thank you for revising the manuscript - this additional work added to the qyaulity of the paper. I have no further comments. I wish you good luck with your work.